# Measurements of Anterior and Posterior Corneal Curvatures with OCT and Scheimpflug Biometers in Patients with Low Total Corneal Astigmatism

**DOI:** 10.3390/jcm11236921

**Published:** 2022-11-24

**Authors:** Maria Muzyka-Woźniak, Adam Oleszko, Andrzej Grzybowski

**Affiliations:** 1Ophthalmology Clinical Center SPEKTRUM, Zaolzianska Str. 4, 53-334 Wroclaw, Poland; 2Department of Ophthalmology, University of Warmia and Mazury, Zołnierska Str. 18, 10-561 Olsztyn, Poland; 3Institute for Research in Ophthalmology, Foundation for Ophthalmology Development, Mickiewicza 24, 61-836 Poznan, Poland

**Keywords:** astigmatism, keratometry, posterior cornea

## Abstract

Background: Posterior keratometry measurements are evolving features of the optical biometers. The differences between devices have bigger impact for the low astigmatism values. The majority of adults present the corneal astigmatism below 1.5 D. Objectives: To compare the total corneal astigmatism measured with two different technologies in cataract patients with corneal astigmatism below 1.5 D. Material and Methods: Three automated exams were performed on each of the two devices: swept-source optical coherence tomography (SS-OCT) and Scheimpflug biometers. The anterior and total corneal astigmatism and power were analysed. Statistical comparisons were performed for within-subject standard deviation, repeatability, Bland–Altman and vector analysis. Results: Twenty-nine eyes of twenty-seven patients were included. The limits of agreement between anterior and total corneal astigmatism were narrower for the SS-OCT than for the Scheimpflug biometer (−0.16 to 0.29 D and −0.40 to 0.39 D, respectively). The >0.5 D difference between SS-OCT and Scheimpflug total astigmatism was noticed in 5 (17%) of cases. The difference between mean total keratometric power for both devices was statistically significant (0.2 D, *p* < 0.001). SS-OCT total corneal flat measurements had worse repeatability than Scheimpflug (*p* = 0.007). Conclusions: For the corneal astigmatism <1.5 D, the difference between anterior and total corneal astigmatism measured with SS-OCT was clinically not significant. The mean anterior and total keratometry values obtained with Scheimpflug and SS-OCT biometers are not interchangeable.

## 1. Introduction

Modern lens-based surgery requires precise intraocular lens (IOL) calculations, including corneal astigmatism correction. Using anterior corneal keratometry only can result in astigmatism over- or under-correction [1]. Incorporating posterior corneal astigmatism into IOL power calculations has been shown to improve the results of toric IOLs [1,2,3,4]. Methods based on theoretical estimation of posterior corneal astigmatism outperform the calculations based on real measurements [3]. This may be caused by technical difficulties in posterior corneal curvature assessment, including both hardware and software issues.

Two main technologies are applied to evaluate the corneal back surface radii: optical coherence tomography (OCT) and Scheimpflug camera. However, each device uses specific manufacturer’s algorithms to derive the radius from the measured raw data. Therefore, the results are often regarded as not interchangeable [5,6,7,8]. For anterior and posterior keratometry measurements, studies have shown significant differences between swept-source OCT (SS-OCT) and Scheimpflug devices [9,10,11,12,13,14,15,16,17], which are considered clinically irrelevant [9,10] or relevant [5,11,12,13,14,15,16,17], depending on interpretation. Specifically, Shajari found a difference of >0.50 diopters between total astigmatism measurements on the SS-OCT biometer and Scheimpflug device in 10 out of 93 subjects [5] and Wang reported 84.3% of eyes with differences of ≤0.50 D and ≤1.0 D in mean astigmatism magnitude on the SS-OCT biometer and Dual–Scheimpflug device [17].

According to UK Biobank Study, only 11% of adults present corneal astigmatism higher than 1.5 D, and the mean value of corneal astigmatism for adults between 40–69 years is close to 0.8 D [18]. Hence, most of the cataract or refractive patients are expected to have low astigmatism. The differences between devices in posterior corneal astigmatism measurements, as well as intra-device repeatability, may have bigger impact for the low than for the high astigmatism, i.e., the difference of ±0.3 D in astigmatism close to 0.7 D influences the decision of implanting a toric IOL, while it is not an issue with an astigmatism close to 3.0 D.

Historically, the Scheimpflug device was the first widely used instrument for posterior corneal measurements. Introducing OCT technology for both axial length and posterior cornea measurements calls into question using a second instrument for total corneal astigmatism assessments for every patient. These measurements, however, directly influence the decision of using or not using a toric IOL. Apart from obvious visual consequences, the financial cost is also a considerable issue.

The purpose of our study was to prospectively assess the total corneal astigmatism derived from direct anterior and posterior corneal curvature measured with an SS-OCT biometer (IOL Master 700; Carl Zeiss Meditec, Jena, Germany) and a Scheimpflug-based biometer (Pentacam AXL, OCULUS Optikgeräte, Wetzlar, Germany) in cataract patients with astigmatism below 1.5 D.

## 2. Methods

The consecutive cataract patients, who came for pre-operative assessments, were enrolled in this prospective controlled study. The exclusion criteria were: keratometric astigmatism above 1.5 D, any corneal pathology, dry eye disease, previous laser corneal refractive surgery, narrow palpebral fissure and retinal disease with macular involvement. If both eyes were eligible, the eye to be operated first was included. The study was carried out in compliance with the tenets of the Declaration of Helsinki, and the research was reviewed and approved by the local ethics committee (No. 77/2021). Prior informed consent was obtained from the subjects. 

### 2.1. Biometric Measurements

All patients underwent 3 automated exams on each of the 2 devices: swept source OCT (SS-OCT) biometer (IOL Master 700, Carl Zeiss Meditec, Jena, Germany) and Scheimpflug camera (Pentacam AXL; OCULUS Optikgeräte, Wetzlar, Germany). The Pentacam AXL is a single rotating Scheimpflug camera device, using the ray-tracing method for the total corneal refractive power (TCRP) evaluation. The radii of curvature for the front and back surfaces of the cornea and the indices of the refraction for air, cornea and aqueous (1.000, 1.376 and 1.336, respectively) are applied. The IOLMaster 700 is a SS-OCT biometer and uses reflectance keratometry, pachymetry and posterior corneal curvature to measure total keratometric power, which is called the Total Keratometry (TK). Using a thick lens formula, the IOLMaster total keratometry astigmatism is calculated using anterior and posterior corneal curvatures and corneal thickness. 

The order of measurements was chosen in a randomised manner. All exams were performed by 2 experienced examiners in the same dim ambient light conditions (0.02 Lux), no longer than 15 min apart. Exams were included only if positively evaluated by automatic quality checks on both devices. 

The following variables were recorded and analysed from the SS-OCT biometer: K_flat_—keratometry in flat corneal meridian; K_steep_—keratometry in steep meridian; K_mean_—mean keratometry; K_ast_—difference between steep and flat meridian representing anterior corneal astigmatism; TK_flat_—total keratometry in flat corneal meridian; TK_steep_—total keratometry in steep meridian; TK_mean_—mean total keratometry; TK_ast_—difference between steep and flat meridian representing total corneal astigmatism. Corresponding parameters were recorded and analysed from the Scheimpflug device: SimK_flat—_simulated keratometry in flat corneal meridian; SimK_steep_—simulated keratometry in steep meridian; SimK_mean_—simulated mean keratometry; SimK_ast_—difference between steep and flat meridian representing anterior corenal astigmatism; TCRP_flat_—total corneal refractive power in flat corneal meridian; TCRP_steep_—total corneal refractive power in steep meridian; TCRP_mean_—mean total corneal refractive power; TCRP_ast_—difference between TCRP_steep_ and TCRP_flat_ representing total corneal refractive astigmatism. Additionally, anterior chamber depth (ACD) defined as the distance between corneal epithelium and anterior surface of crystalline lens apex and central corneal thickness (CCT) were recorded.

### 2.2. Corneal Astigmatism Assessment

For the astigmatism assessment, the vector analysis was performed, according to Thibos [19]. The following values were used: anterior (K_steep_−K_flat_) and total (TK_steep_−TK_flat_) astigmatism for the SS-OCT device and anterior (SimK_steep_−SimK_flat_) and TCRP astigmatism (TCRP_steep_−TCRP_flat_) for the Scheimflug device. These were converted to rectangular vectors J0 and J45 using the following equations: J0 = −(C/2) × cos (2α) and J45 = −(C/2) × sin (2α), where J0 represents the Jackson cross-cylinder power at axis 90° and 180°, J45 is the Jackson cross-cylinder power at axis 45° and 135°, C is the astigmatism magnitude, and α is the axis of the astigmatism. In this notation, positive J0 indicates with-the-rule (WTR) astigmatism and negative indicates against-the-rule (ATR) astigmatism. J45 represents oblique astigmatism. Double angle plots were created. 

A predefined clinically significant limit of 0.5 D difference between anterior and total astigmatism and between TK astigmatism and TCRP astigmatism was assessed, as described by Shajari [5]. 

### 2.3. Statistics

Sample size calculation was performed according to McAlinden and at least 27 subjects to achieve 0.25 D of astigmatism limit of agreement were considered [20]. Data were recorded using Excel (Microsoft Corp., Redmond, WA, USA) and presented as mean ± standard deviation (SD). 

Matlab R2009b software (MathWorks, Inc., Natick, MA, USA) was applied to perform statistical analysis. Normal distribution of all data sets was evaluated with the Kolmogorov–Smirnov test. Statistical significance of differences between measurements from the two devices was assessed with a paired sample *t*-test. For statistical analyses, a *p*-value less than 0.05 was considered significant. Agreement between devices was evaluated with Bland–Altman plots. Origin 8.0 software (OriginLab, Co., Northampton, MA, USA) was applied to create the plots. For the double angle plots, the ASCRS tool (https://ascrs.org/tools/astigmatism-double-angle-plot-tool/, accessed on 10 September 2022) was used [21].

The repeatability of both devices was evaluated according to Bland and Altman [22]. The within-subject SD (Sw) was based on 3 measurements of the same eye. The coefficient of variance (CoV) was calculated as the ratio of Sw to the mean. The test–retest 95% confidence interval (CI) was calculated [23]. For 95% CI of the limits of agreement (LoA), the exact method was used [24].

## 3. Results

This prospective study included twenty-nine eligible eyes (17 right, 12 left eyes) of 27 patients (7 men, 20 women), with a mean age of 68 ± 11 years (range 40–84). For two subjects, both eyes were used because the measurements for each eye were performed separately, with a >3 month time interval. Mean axial length was 23.26 ± 1.02 mm (range 20.84–25.71). The mean K_ast_ was 0.72 ± 0.36 D and the mean SimK_ast_ was 0.74 ± 0.41 D (*p* = 0.46, *t*-test). Other corresponding parameters measured on the two devices are presented in Table 1.

The S_w_ for all variables was comparable, with the only statistical significant difference between TK_flat_ (Sw = 0.12) and TCRP_flat_ (Sw = 0.07, *p* = 0.007) (Table 2).

The mean difference between mean K and mean TK values measured with an SS-OCT biometer was statistically not significant. The mean difference between keratometric corneal astigmatism and total keratometric corneal astigmatism was statistically significant, with a mean difference of 0.07 D, which is not clinically significant, and narrow limits of agreement (Figure 1)

The mean difference between simK_mean_ and TCRP_mean_ was statistically significant (95% LoA −0.42–0.70), while the difference between simK_ast_ and TCRP_ast_ was not statistically significant (Figure 2).

We observed no significant differences between K and simK values, except the steep meridian. The mean differences between TK and TCRP for flat, steep and mean values were statistically significant. The mean difference between TK_ast_ and TCRP_ast_ was not statistically significant (*p* = 0.379) (Figure 3). Differences between measured indices within and between devices are presented in Table 3.

The difference between anterior and total astigmatism did not exceed a predefined clinically significant limit of 0.5 D for SS-OCT biometer in any of the analysed eyes. For the Scheimpflug SimK_ast_ and TCRP_ast_, the 0.5 D difference was exceeded in 2 (7%) cases out of 29. The >0.5 D difference between TK_ast_ and TCRP_ast_ was noticed in 5 (17%) cases out of 29.

The power vector analysis is shown in Table 4 and Figure 4a–d. There were no statistically significant differences in vector analysis of anterior and total corneal astigmatism measured on both devices, but wide limits of agreement range were noticed. Figure 5 shows double angle plots for K, SimK, TK and TCRP.

## 4. Discussion

We showed that, for cataract patients with anterior corneal astigmatism <1.5 D, the difference between anterior and total corneal astigmatism measured with SS-OCT and Scheimpflug biometers was not clinically significant. However, the 95% LoA were wider for Scheimpflug (0.8 D) than for SS-OCT (0.44 D), while both devices showed excellent repeatability. The >0.5 D difference between TK astigmatism and TCRP astigmatism was noticed in 17% of cases. Therefore, the results of total corneal astigmatism measurements obtained with these two different technologies cannot be regarded as interchangeable.

The posterior cornea curvature is difficult to measure due to a small difference in refractive indexes between posterior cornea (1.367) and the aqueous humor (1.336). The examined devices use different technology for posterior corneal surface evaluation and the validation of the measurements may be different as well. Scheimpflug-based technology uses ray tracing for TCRP estimation, with parallel light beams refraction, slope and location. The SS-OCT biometer combines reflectance keratometry, pachymetry and posterior corneal curvature to calculate the TK value. We found a wide range of agreement between TK and TCRP astigmatism (1.15 D), although the vector analysis did not show significant differences between the devices, with a mean difference of 0.08 (J0) and 0.01 (J45). Shajari et al. reported a bigger difference in an astigmatism vector for TK and TCRP comparison (0.15 ± 0.36 at 12); the authors also reported that better agreement was achieved for TK and True Net Power (TNP) astigmatism comparison [5]. The TNP represents the optical power derived from sagittal curvatures of both corneal surfaces and correct refractive indices. Other authors consider equivalent keratometric readings, which are derived from corneal elevation values of the anterior and posterior surfaces in addition to local corneal thickness, as more appropriate for comparing Pentacam system with other devices [6]. The divergence in parameters chosen for comparisons shows clearly that there is no clinical gold standard for the total corneal astigmatism measures.

Savini et al. presented more detailed analyses of total corneal astigmatism measured with TK and Scheimpflug, separately for eyes with with-the-rule, against-the rule and oblique astigmatism at 2 mm and 3 mm zone/apex and ring/apex. The only statistically significant difference between devices was present for the polar value along the 45-degree meridian in against-the rule and with-the rule eyes [8]. In our group, limited to 1.5 D of astigmatism, no significant differences in vectorial values were found.

Shajari et al. proposed testing the differences in total corneal astigmatism measurements exceeding 0.50 D (a clinically significant limit), which may be concealed in a calculation for the mean difference, as in the Bland–Altman analysis [5]. They found more eyes exceeding this limit between simK and TCRP astigmatism than with K and TK astigmatism. Our data support these results. The difference between K and TK astigmatism did not exceed a limit of 0.5 D in any of the analysed eyes (maximal difference 0.33 D). Savini and co-authors found a difference of more than 0.5 D between keratometric and total corneal astigmatism in 16.6% of eyes with 1.00 D or more of corneal astigmatism, measured with a Scheimpflug device [1]. Since the magnitude of posterior astigmatism is not a constant value and may increase with the magnitude of anterior with-the-rule astigmatism, one can expect larger keratometric and total astigmatism differences variation with larger astigmatism values [25].

The comparison of mean K and TK showed narrow LoA and bias line close to zero. The mean difference between these two variables was neither statistically nor clinically significant (0.02 D), which is in agreement with other studies [1,5]. The IOLMaster 700 TK values are originally adjusted by the manufacturer (Carl Zeiss Meditec, Jena, Germany) to be used with usual IOL constants and calculation formulas for IOL spherical equivalent estimation. The Scheimpflug device TCRP was significantly different from simK values and should not be used in IOL calculation formulas, due to the double compensation for the refraction indices: one in the formula, second in the TCRP by definition. Therefore, TK and TCRP cannot be regarded as comparable.

The mean dioptric corneal power in our cohort was significantly lower for TCRP by 0.2 D in comparison to the TK value. These results are in agreement with other reports [5,6]. Therefore, for toric IOL calculations, significant differences can be expected, if real total corneal power values from different devices would be applied.

The direct measurement of posterior corneal surface offers clear benefit for the IOL calculation in eyes after previous keratorefractive surgery. Lawless showed that using TK in the Barrett True-K formula yields lower mean refractive prediction error in such patients [26].

The main limitation of the present study is the low number of eyes included in the analysis. We tended to use one eye per patient to offset any interocular dependency issues. Further studies with a larger sample size should follow. The second major limitation is that the compared TK and TCRP are not exactly the same corresponding parameters, not only due to different measurement technology. The TK value is originally adjusted by the posterior corneal power difference from normal. The device manufacturer does not specify the method used for this compensation. Therefore, we assume that, without the adjustment, the results might be different. However, comparing total corneal astigmatism from both devices is justified because TK_ast_ astigmatism and TCRP_ast_ are the vector sum of anterior and posterior astigmatism. Another major limitation of the present study is that similar comparisons were already published, but our specific limitation to low astigmatism values exposes the difficulty in assessments used as cut-offs for surgical method selection in such cases.

## 5. Conclusions

Regarding low corneal astigmatism (<1.5 D), the difference between measured anterior and total corneal astigmatism was not clinically significant, when assessed with IOLMaster 700. Wider LoA between anterior and total corneal astigmatism were noticed for the Pentacam AXL. Both technologies evaluate the posterior corneal curvature with good repeatability, but the results cannot be regarded as interchangeable.

## Figures and Tables

**Figure 1 jcm-11-06921-f001:**
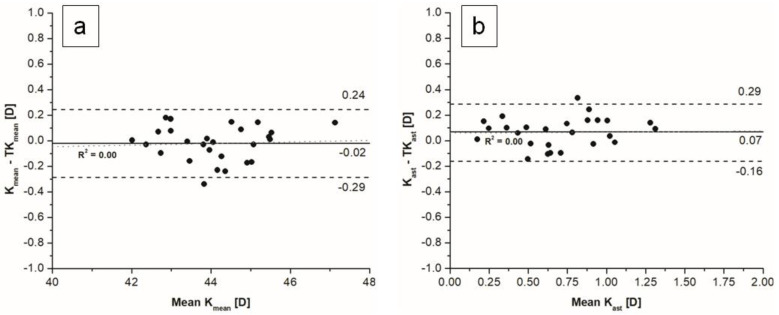
Bland–Altman plots of comparison between K and TK using (**a**) mean values, (**b**) astigmatism magnitude.

**Figure 2 jcm-11-06921-f002:**
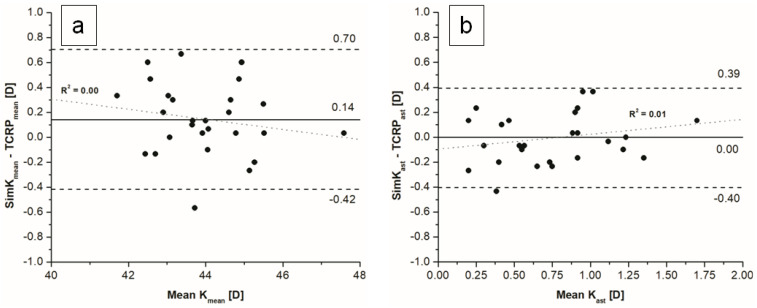
Bland–Altman plots of comparison between simK and TCRP using (**a**) mean values, (**b**) astigmatism magnitude.

**Figure 3 jcm-11-06921-f003:**
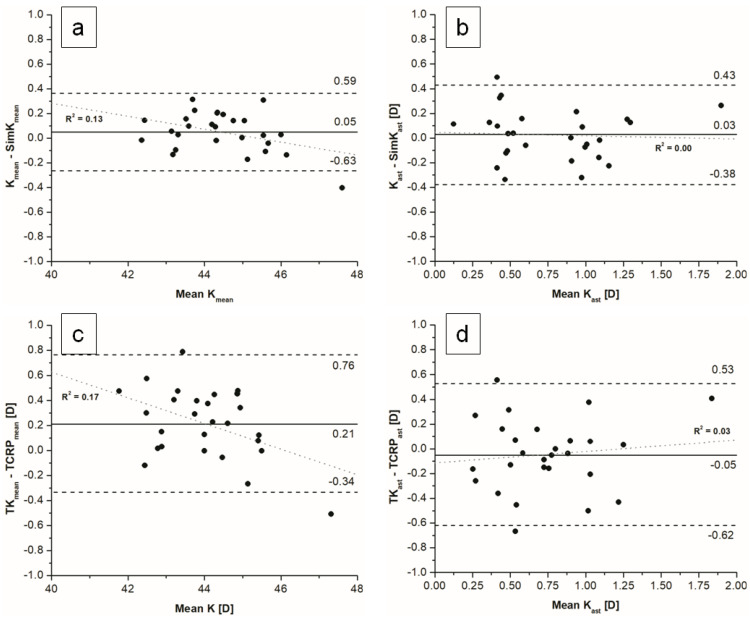
Bland–Altman plots of comparison between K and simK using (**a**) mean values, (**b**) astigmatism magnitude and for TK and TCRP using (**c**) mean values, (**d**) astigmatism magnitude.

**Figure 4 jcm-11-06921-f004:**
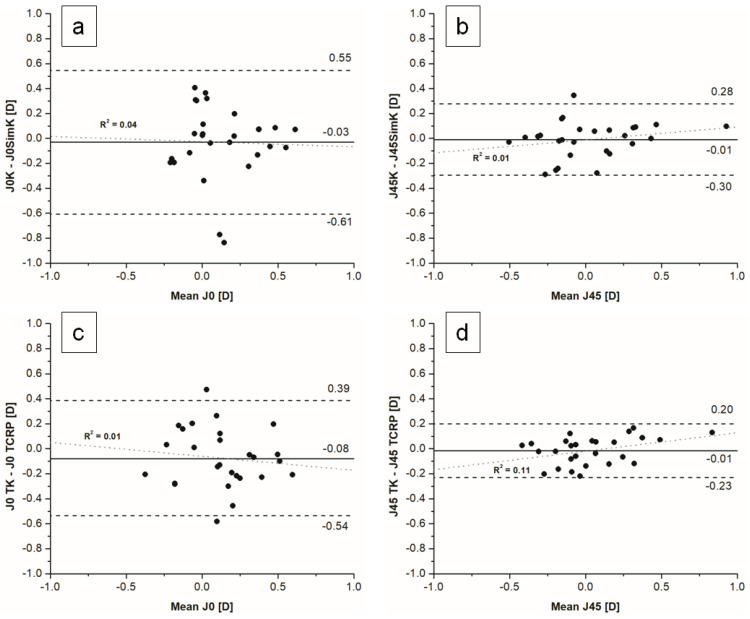
Bland–Altman plots for astigmatism vector analysis for K and simK (**a**) J0, (**b**) J45 and for TK and TCRP (**c**) J0, (**d**) J45.

**Figure 5 jcm-11-06921-f005:**
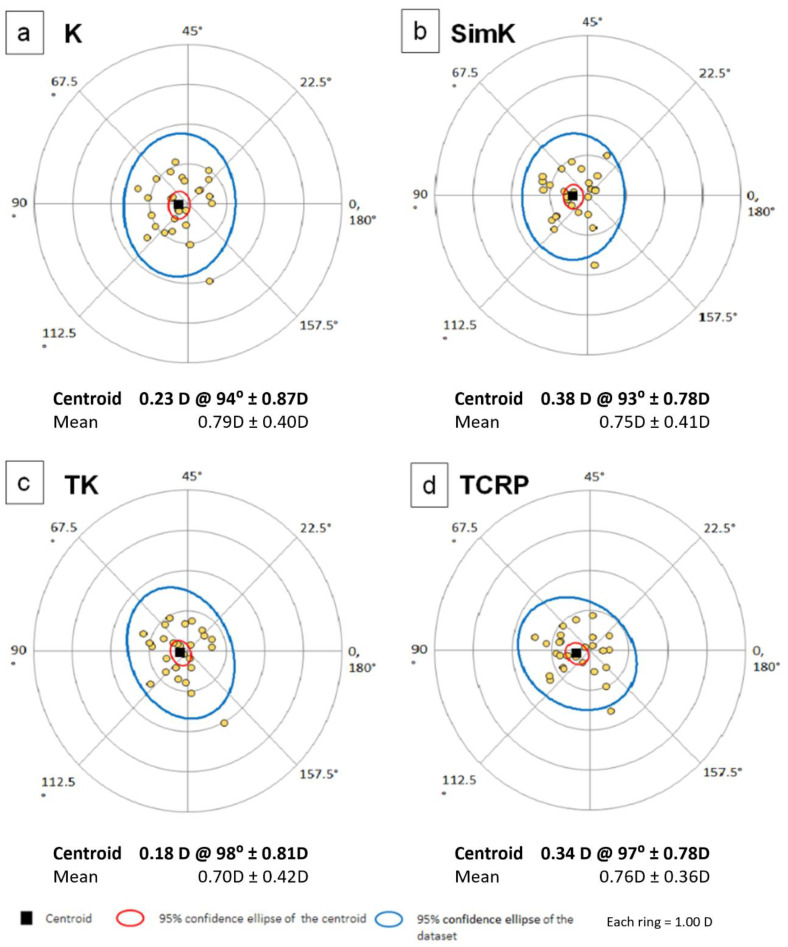
Double-angle plots showing the astigmatism measured with (**a**) K, (**b**) simK, (**c**) TK, (**d**) TCRP. K—keratometry; simK—simulated keratometry; TK—total keratometry, TCRP—total corneal refractive power.

**Table 1 jcm-11-06921-t001:** Corresponding variables measured with two devices.

IOL Master 700					Pentacam AXL		
Parameter	Mean ± SD	Range	95% CI	Parameter	Mean ± SD	Range	95% CI
K_flat_ [D]	43.71 ± 1.23	41.52–47.05	(0.13–0.84)	SimK_flat_ [D]	43.66± 1.27	41.33–47.33	(0.11–0.70)
K_steep_ [D]	44.50 ± 1.17	42.50–47.34	(0.09–0.55)	SimK_steep_ [D]	44.42 ± 1.26	42.37–47.87	(0.14–0.19)
K_mean_ [D]	44.10 ± 1.18	42.01–47.20	(0.09–0.56)	SimK_mean_ [D]	44.05 ± 1.25	41.87–47.60	(0.11–0.71)
K_ast_ [D]	0.72 ± 0.36	0.18–1.37	(0.15–0.96)	SimK_ast_ [D]	0.74 ± 0.41	0.07–1.8	(0.12–0.77)
TK_flat_ [D]	43.78 ± 1.22	41.69–46.99	(0.17–1.06)	TCRP_flat_ [D]	43.52 ± 1.34	41.13–47.37	(0.08–0.49)
TK_steep_ [D]	44.48 ± 1.15	42.33–47.13	(0.09–0.57)	TCRP_steep_ [D]	44.28 ± 1.28	41.93–47.77	(0.11–0.71)
TK_mean_ [D]	44.12 ± 1.17	42.01–47.06	(0.07–0.45)	TCRP_mean_ [D]	43.90 ± 1.30	41.53–47.57	(0.07–0.45)
TK_ast_ [D]	0.64 ± 0.31	0.14–1.27	(0.15–0.97)	TCRP_ast_ [D]	0.76 ± 0.35	0.13–1.63	(0.13–0.84)
ACD [mm]	3.01 ± 0.39	2.29–3.81	(0.00–0.03)	ACD [mm]	3.03 ± 0.39	2.31–3.85	(0.01–0.04)
CCT [µm]	553 ± 27	501–606	(2–12)	CCT [µm]	543 ± 26	489–593	(2–15)

K_flat_—keratometry in flat corneal meridian; K_steep_—keratometry in steep meridian; K_mean_—mean keratometry; K_ast_—anterior corneal astigmatism; TK_flat_—total keratometry in flat corneal meridian; TK_steep_—total keratometry in steep meridian; TK_mean_—mean total keratometry; TK_ast_—total corneal astigmatism; SimK_flat—_simulated keratometry in flat corneal meridian; SimK_steep_—simulated keratometry in steep meridian; SimK_mean_—simulated mean keratometry; SimK_ast_—simulated anterior corenal astigmatism; TCRP_flat_—total corneal refractive power in flat corneal meridian; TCRP_steep_—total corneal refractive power in steep meridian; TCRP_mean_—mean total corneal refractive power; TCRP_ast_—total corneal refractive astigmatism; ACD—anterior chamber depth; CCT—central corneal thickness; SD—standard deviation; 95%CI—within subject test–retest repeatability 95% confidence interval.

**Table 2 jcm-11-06921-t002:** Comparison of the repeatability of the measured variables.

IOL Master 700			Pentacam AXL			Master vs. Pentacam Repeatability
Variable	S_w_	CoV [%]	Variable	S_w_	CoV [%]	*p*-Value
K_flat_ [D]	0.10	0.2	SimK_flat_ [D]	0.10	0.2	0.626
K_steep_ [D]	0.07	0.2	SimK_steep_ [D]	0.11	0.2	0.153
K_mean_ [D]	0.07	0.2	SimK_mean_ [D]	0.09	0.2	0.628
K_ast_ [D]	0.11	18.2	SimK_ast_ [D]	0.10	20.4	0.336
TK_flat_ [D]	0.12	0.3	TCRP_flat_ [D]	0.07	0.2	0.007 *
TK_steep_ [D]	0.08	0.2	TCRP_steep_ [D]	0.10	0.2	0.595
TK_mean_[D]	0.07	0.2	TCRP_mean_ [D]	0.06	0.1	0.135
TK_ast_ [D]	0.12	25.8	TCRP_ast_ [D]	0.12	18.9	0.682
ACD [mm]	0.00	0.1	ACD [mm]	0.01	0.2	1.000
CCT [µm]	2	0.3	CCT [µm]	2	0.4	0.181

Statistically significant *p*-values (<0.05) are marked with asterisk (paired *t*-test). K_flat_—keratometry in flat corneal meridian; K_steep_—keratometry in steep meridian; K_mean—_mean keratometry; K_ast_—anterior corneal astigmatism; TK_flat_—total keratometry in flat corneal meridian; TK_steep_—total keratometry in steep meridian; TK_mean_—mean total keratometry; TK_ast_—total corneal astigmatism; SimK_flat—_simulated keratometry in flat corneal meridian; SimK_steep_—simulated keratometry in steep meridian; SimK_mean_—simulated mean keratometry; SimK_ast_—simulated anterior corenal astigmatism; TCRP_flat_—total corneal refractive power in flat corneal meridian; TCRP_steep_—total corneal refractive power in steep meridian; TCRP_mean_—mean total corneal refractive power; TCRP_ast_—total corneal refractive astigmatism; ACD—anterior chamber depth; CCT—central corneal thickness; S_w_—within-subject standard deviation; CoV—coefficient of variance.

**Table 3 jcm-11-06921-t003:** Comparison of the IOL Master 700 and Pentacam AXL indices.

Indices (D)	*p*-Value	Mean Difference	Min Difference	Max Difference	Upper LoA (95%CI)	Lower LoA (95%CI)	LoA Range	R^2^
IOL Master 700								
K_mean_ vs. TK_mean_	0.465	−0.02	−0.34	0.18	0.24 (0.15, 0.33)	−0.29 (−0.38, −0.20)	0.53	0.00
K_ast_ vs. TK_ast_	0.008 *	0.07	−0.14	0.33	0.29 (0.22, 0.36)	−0.16 (−0.23, −0.09)	0.44	0.00
Pentacam AXL								
SimK_mean_ vs. TCRP_mean_	0.019 *	0.14	−0.57	0.67	0.70 (0.50, 0.90)	−0.42 (−0.62, −0.22)	1.12	0.00
SimK_ast_ vs. TCRP_ast_	0.437	0.01	−0.43	0.37	0.39 (−0.26, 0.53)	−0.40 (−0.54, −0.26)	0.80	0.01
IOL Master vs. Pentacam AXL								
K_flat_ vs. SimK_flat_	0.227	0.05	−0.40	0.39	0.47 (−0.22, 0.62)	−0.37 (−0.52, −0.22)	0.84	0.04
K_steep_ vs. SimK_steep_	0.027 *	0.09	−0.52	0.45	0.49 (−0.17, 0.63)	−0.31 (−0.45, −0.17)	0.80	0.22
K_mean_ vs. SimK_mean_	0.117	0.05	−0.40	0.31	0.36 (−0.16, 0.47)	−0.27 (−0.38, −0.16)	0.63	0.13
K_ast_ vs. SimK_ast_	0.465	0.03	−0.34	0.49	0.43 (0.16, 0.97)	−0.38 (−0.92, 0.16)	0.81	0.00
TK_flat_ vs. TCRP_flat_	<0.001 *	0.26	−0.57	0.89	0.92 (−0.17, 1.15)	−0.40 (−0.63, −0.17)	1.32	0.12
TK_steep_ vs. TCRP_steep_	0.002 *	0.20	−0.64	0.76	0.79 (−0.18, 0.99)	−0.38 (−0.58, −0.18)	1.17	0.21
TK_mean_ vs. TCRP_mean_	<0.001 *	0.21	−0.51	0.79	0.76 (−0.14, 0.95)	−0.33 (−0.52, −0.14)	1.10	0.20
TK_ast_ vs. TCRP_ast_	0.379	−0.05	−0.67	0.56	0.53 (−0.13, 1.02)	−0.62 (−1.11, −0.13)	1.15	0.01

Statistically significant *p*-values (<0.05) are marked with asterisk (paired *t*-test). K_flat_—keratometry in flat corneal meridian; K_steep_—keratometry in steep meridian; K_mean_—mean keratometry; K_ast_—anterior corneal astigmatism; TK_flat_—total keratometry in flat corneal meridian; TK_steep_—total keratometry in steep meridian; TK_mean_—mean total keratometry; TK_ast_—total corneal astigmatism; SimK_flat—_simulated keratometry in flat corneal meridian; SimK_steep_—simulated keratometry in steep meridian; SimK_mean_—simulated mean keratometry; SimK_ast_—simulated anterior corenal astigmatism; TCRP_flat_—total corneal refractive power in flat corneal meridian; TCRP_steep_—total corneal refractive power in steep meridian; TCRP_mean_—mean total corneal refractive power; TCRP_ast_—total corneal refractive astigmatism.; LoA—limits of agreement; CI—confidence interval.

**Table 4 jcm-11-06921-t004:** The vector analysis of the anterior and total corneal astigmatism obtained on IOLMaster700 and Pentacam AXL.

IOL Master 700			Pentacam AXL			Difference		
(D)	Mean ± SD	Range		Mean ± SD	Range	Mean Difference	95% LoA Range	*p*
J0_K_	0.11 ± 0.49	−1.52–1.49	J0_SimK_	0.13 ± 0.47	−1.44–1.49	−0.03	1.15	0.596
J45_K_	0.04 ± 0.34	−0.41–0.98	J45_SimK_	0.05 ± 0.32	−0.49–0.88	−0.01	0.57	0.732
J0_TK_	0.07 ± 0.48	−1.60–1.42	J0_TCRP_	0.15 ± 0.50	−1.58–1.55	−0.08	0.92	0.082
J45_TK_	0.04 ± 0.32	−0.41–0.90	J45_TCRP_	0.07 ± 0.28	−0.43–0.77	−0.01	0.43	0.514

J0—Jackson cross-cylinder power at axis 90° and 180°; J45—Jackson cross-cylinder power at axis 45° and 135°, K—keratometry; TK—total keratometry; simK—simulated keratometry; TCRP—total corneal refractive power; SD—standard deviation.

## Data Availability

The data presented in this study are available on request from the corresponding author. The data are not publicly available due to General Data Protection Regulation at the Institution.

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
