# Peer review of "Measurements of Anterior and Posterior Corneal Curvatures with OCT and Scheimpflug Biometers in Patients with Low Total Corneal Astigmatism"

_jcm, 2022, doi:10.3390/jcm11236921_

Round 1

Reviewer 1 Report

In this paper, authors compare two methods of measuring corneal curvatures, SS-OCT and Scheimpflug imaging, in a low total corneal astigmatism scenario.

The article is well written, the experiment design is ok, and English is good enough to understand the paper without effort.

 The results are clearly presented, and conclusions are supported by the results.

 However, the topic of the article has been extensely investigated, and many similar articles comparing these technologies in different scenarios have been published, so the article suffers a bit from lack of novelty.

 Also, the small sample size makes difficult to extrapolate this research conclusions to greater populations, as there could be unnoticed biases present. Repeating the experiment with a bigger sample would make it much more interesting.

Here are some additional comments that may help to improve the manuscript before its publication:

 Introduction - There are many references about this topic that are missing, I would recommend to include most important previous studies that compare SS-OCT and Scheimpflug imaging in terms of keratometry. Also, previous state of the art on technologies interchangeability must be mentioned and cited.

 Table 1 - Please include SD meaning in footnotes of the table.

 Table 2 - Please include Sw and CoV meaning in footnotes of the table.

 Line 280 - Please complete the sentence "Another major limitation of the present study..."

 References - The number of cited references is a bit low for a 10 page paper, I would recommend to increase it to at least 30 sources or more.

Author Response

Thank you for the valuable review of our manuscript. We were happy to improve our work accordingly.

In this paper, authors compare two methods of measuring corneal curvatures, SS-OCT and Scheimpflug imaging, in a low total corneal astigmatism scenario.

 The article is well written, the experiment design is ok, and English is good enough to understand the paper without effort.

 The results are clearly presented, and conclusions are supported by the results.

 However, the topic of the article has been extensely investigated, and many similar articles comparing these technologies in different scenarios have been published, so the article suffers a bit from lack of novelty.

 Also, the small sample size makes difficult to extrapolate this research conclusions to greater populations, as there could be unnoticed biases present. Repeating the experiment with a bigger sample would make it much more interesting.

Response: We are aware that one of the major limitations of out study is that similar comparisons were already published, but our specific limitation to low astigmatism values exposes the difficulty in assessments used as cut-offs for surgical method selection in such cases, i.e. choosing the low toric intraocular lens (T2) or on-axis main incision placement. We think that showing the challenges of measuring low corneal astigmatism is important for surgeons implanting multifocal or extended depth of field intraocular lenses. Obviously, the small sample size is another important limitation of this study and bigger sample should follow.  We included these statements in Discussion.

 Introduction - There are many references about this topic that are missing, I would recommend to include most important previous studies that compare SS-OCT and Scheimpflug imaging in terms of keratometry. Also, previous state of the art on technologies interchangeability must be mentioned and cited.

Response: We have updated the references and included them in the introduction. 

 Table 1 - Please include SD meaning in footnotes of the table.

Response: This is was corrected

 Table 2 - Please include Sw and CoV meaning in footnotes of the table.

Response: This is was corrected

 Line 280 - Please complete the sentence "Another major limitation of the present study..."

Response: Another major limitation of the present study is that similar comparisons were already published, but our specific limitation to low astigmatism values exposes the difficulty in assessments used as cut-offs for surgical method selection in such cases.

 References - The number of cited references is a bit low for a 10 page paper, I would recommend to increase it to at least 30 sources or more.

Response: The most relevant references were added.

Reviewer 2 Report

Review: Measurements of anterior and posterior corneal curvatures with OCT and Scheimpflug biometers in patients with low total corneal astigmatism

General comment.

The research compares astigmatism outputs values and repeatability from IOL Master 700 and Pentacam AXL; OCULUS. Generally, this draft is well written. Despite not being a novel topic, it helps add evidence to astigmatism's studies.

Specific comments.

Line 49: It needs to be clarified what the authors mean by "bigger impact"

Line 99: Replace "crystalline lens" with "crystalline lens apex"

Line 103: Repair Thibos reference

Line 126: What software was used for creating the Double angle plots?

Line 130: Some comments about statistics:

Authors could considerably improve the information given if the 95% CI for the TRT were calculated (e.g. Barnhart & Barboriak, 2009) and the 95% CI of the LoA calculated using the exact methods (Carkeet 2015).

Line 132: Why did two subjects contribute with two yes? It needs to be justified.

Line 134: Replace "23.26 mm ±1.02" with "23.26 ±1.02 mm"

Line 141-147: Caption of Table 1. Replace "=" with "-" and do the same to the other tables.

Line 152 (Table 2): 

Sw and test-retest repeatability are related, with test-retest repeatability being approximately 1.96*(2^.5) *Sw. So, there is little value in reporting both.

Line 175: Repair "statisitcally"

Line 189-190 (Caption Table 3): delete "ACD- anterior chamber depth; CCT = 189 central corneal thickness"

Line 207: Repair "keratemetry"

Line 227: Repair "((0,15"

Line 280: "Another major limitation of the present study" seems to be a loose statement.

Regarding limitations: This study compares two specific instruments. An unstated assumption is that these instruments are representative of their types. That assumption should be recognized.

Author Response

Thank you for the valuable review of our manuscript. We were happy to improve our work accordingly.

 General comment.

The research compares astigmatism outputs values and repeatability from IOL Master 700 and Pentacam AXL; OCULUS. Generally, this draft is well written. Despite not being a novel topic, it helps add evidence to astigmatism studies.

Specific comments.

Line 49: It needs to be clarified what the authors mean by "bigger impact"

Response: The differences between devices in posterior corneal astigmatism measurements, as well as intra-device repeatability, may have a bigger impact for the low than for high astigmatism, i.e. the difference of ±0.3 D in astigmatism close to 0.7 D influences the decision of implanting a toric IOL, while it is not an issue with astigmatism close to 3.0 D.

Line 99: Replace "crystalline lens" with "crystalline lens apex"

Response: This was corrected

Line 103: Repair Thibos reference

Response: This was corrected

Line 126: What software was used for creating the Double angle plots?

Response:

An Excel tool was used to create double-angle plots. The tool was obtained from ASCRS webpage: https://ascrs.org/tools/astigmatism-double-angle-plot-tool/

Calculations were performed following Abulafia et al. 2018.

Abulafia A, Koch DD, Holladay JT, Wang L, Hill WE. Editorial. Pursuing Perfection in IOL Calculations IV: Astigmatism analysis, SIA and double angle plots. Journal of Cataract and Refractive Surgery. 2018; 44(10): 1169 - 1174.

Line 130: Some comments about statistics:

Authors could considerably improve the information given if the 95% CI for the TRT were calculated (e.g. Barnhart & Barboriak, 2009) and the 95% CI of the LoA calculated using the exact methods (Carkeet 2015).

Response: Data on 95% CI for the TRT were added to the converted Table 1. Data on 95% CI of the LoA calculated using the exact method were added to the converted Table 3. An appropriate description with references was also added to the Statistics section.

Line 132: Why did two subjects contribute with two yes? It needs to be justified.

Response: We have included consecutive eligible patients. For these two subjects, both eyes were used, because the measurements for each eye were performed separately, with > 3 months’ time intervals.

Line 134: Replace "23.26 mm ±1.02" with "23.26 ±1.02 mm"

Response: This was corrected.

Line 141-147: Caption of Table 1. Replace "=" with "-" and do the same to the other tables.

Response: This was corrected

Line 152 (Table 2): 

Sw and test-retest repeatability are related, with test-retest repeatability being approximately 1.96*(2^.5) *Sw. So, there is little value in reporting both.

 Repeatability columns were deleted from Table 2 as they were redundant.

Line 175: Repair "statisitcally"

Response: This was corrected

Line 189-190 (Caption Table 3): delete "ACD- anterior chamber depth; CCT = 189 central corneal thickness"

Response: This was deleted.

Line 207: Repair "keratemetry"

Response: This was corrected

Line 227: Repair "((0,15"

Response: This was corrected

Line 280: "Another major limitation of the present study" seems to be a loose statement.

Response: Another major limitation of the present study is that similar comparisons were already published, but our specific limitation to low astigmatism values exposes the difficulty in assessments used as cut-offs for surgical method selection in such cases.

Regarding limitations: This study compares two specific instruments. An unstated assumption is that these instruments are representative of their types. That assumption should be recognized.

Response: We have added the exact names of the devices in the introduction.

Round 2

Reviewer 1 Report

All my previous comments have been addressed.